# Antibiotic Consumption in a Cohort of Hospitalized Adults with Viral Respiratory Tract Infection

**DOI:** 10.3390/antibiotics12040788

**Published:** 2023-04-20

**Authors:** Sara Debes, Jon Birger Haug, Birgitte Freiesleben De Blasio, Jonas Christoffer Lindstrøm, Christine Monceyron Jonassen, Susanne Gjeruldsen Dudman

**Affiliations:** 1Center for Laboratory Medicine, Østfold Hospital Trust Kalnes, 1714 Grålum, Norway; 2Institute of Clinical Medicine, Faculty of Medicine, University of Oslo, 0372 Oslo, Norway; 3Department of Infection Control, Østfold Hospital Trust Kalnes, 1714 Grålum, Norway; 4Division of Infection Control and Environmental Health, Department of Methods Development and Analytics, Norwegian Institute of Public Health, 0213 Oslo, Norway; 5Institute of Basic Medical Sciences, Department of Biostatistics, Centre for Biostatistics and Epidemiology, University of Oslo, 0372 Oslo, Norway; 6Department of Virology, Norwegian Institute of Public Health, 0213 Oslo, Norway; 7Department of Microbiology, Oslo University Hospital, 0372 Oslo, Norway

**Keywords:** viral respiratory tract infection, hospitalized, adults, microbiology, antibiotic, decision-making, influenza, RSV

## Abstract

Development of antibiotic resistance, a threat to global health, is driven by inappropriate antibiotic usage. Respiratory tract infections (RTIs) are frequently treated empirically with antibiotics, despite the fact that a majority of the infections are caused by viruses. The purpose of this study was to determine the prevalence of antibiotic treatment in hospitalized adults with viral RTIs, and to investigate factors influencing the antibiotic decision-making. We conducted a retrospective observational study of patients ≥ 18 years, hospitalized in 2015–2018 with viral RTIs. Microbiological data were taken from the laboratory information system and information on antibiotic treatment drawn from the hospital records. To investigate decisions for prescribing antibiotic treatment, we evaluated relevant factors such as laboratory and radiological results, in addition to clinical signs. In 951 cases without secondary bacterial RTIs (median age 73 years, 53% female), 720 (76%) were prescribed antibiotic treatment, most frequently beta-lactamase-sensitive penicillins, but cephalosporins were prescribed as first-line in 16% of the cases. The median length of treatment (LOT) in the patients treated with antibiotics was seven days. Patients treated with antibiotics had an average of two days longer hospital stay compared to patients with no such treatment, but no difference in mortality was found. Our study revealed that there is still a role for antimicrobial stewardship to further improve antibiotic use in patients admitted for viral RTIs in a country with relatively low antibiotic consumption.

## 1. Introduction

The development of antibiotic resistance is an increasing global health threat fueled by inappropriate antibiotic usage within and outside health care institutions. The prevalence of multidrug-resistant bacteria has been reported as increasing and spreading worldwide [1,2,3]. According to WHO’s list of threats to global health, antimicrobial resistance is among the top 10 issues listed [4]. WHO has also stated that “systematic misuse and overuse of these drugs in human medicine and food production have put every nation at risk” [5]. Furthermore, antibiotics often cause adverse effects such as the development of *Clostridioides difficile* infections, as well as toxic, allergic and other complications. Frequent and prolonged antimicrobial therapy is associated with higher rates of adverse events and promotes antibiotic resistance development [6,7,8].

Norway is a low-incidence country with regards to the consumption of antibiotics both in the community and in the hospital sector compared to the EU/EEA average. However, resistance to broad-spectrum antibiotics such as third-generation cephalosporins has been increasing over the past ten years [9]. Penicillins (J01C) represent 46% of the use measured in DDD (Defined Daily Dose) in Norwegian hospitals, and the second largest group is cephalosporins, 19% of all DDDs, with the dominant subgroup being third generation cephalosporins (J01DD) [10]. Broad spectrum antibiotics (defined as J01_CR/DC/DD/DI/DF/DH/MA) accounted for 21% of DDDs in 2021 [10].

Inappropriate antibiotic use is driven largely by lack of adherence to guidelines supporting sound antibiotic prescription. In addition, there are shortcomings of appropriate tools for decision-making in antibiotic treatment that is most often initiated by using an empiric regimen to control infections rapidly before results from microbiological investigations are available. Delays in laboratory testing and access to such services can be part of the problem.

The most common viral agents in respiratory tract infections (RTIs) are the influenza virus, respiratory syncytial virus (RSV), human metapneumovirus (hMPV), rhinovirus, parainfluenza virus, adenovirus and coronavirus [11,12]. Bacterial RTIs are caused by *Streptococcus pneumonia*, *Haemophilus influenzae*, *Moraxella catarrhalis*, *Mycoplasma pneumoniae* and Gram-negative bacilli, among others [13,14,15].

Respiratory infections are often treated empirically with antibiotics in primary health care and in hospital settings. However, more widespread use of molecular microbiological tests has revealed that most respiratory tract infections (RTIs) are caused by viruses, and do not require antibiotic treatment [11,16]. Nevertheless, antibiotics are often initiated on clinical suspicion in cases of viral RTIs before the diagnosis is confirmed. Often, clinicians do not discontinue the antibiotic course even after receiving a positive respiratory virus result but no evidence of a bacterial cause of infection [7].

In Norway, the standard care for adults requiring hospital admission due to viral RTIs [17] includes intravenous fluids, antipyretic medicines and oxygen in case of hypoxemia. Hospitalized cases developing critical illness will be evaluated for the need of mechanical ventilation or extracorporeal membrane oxygenation. Antiviral treatment with a neuraminidase inhibitor, such as oseltamivir, will be considered for patients with confirmed or suspected influenza. In addition, antibiotic treatment will be administered to patients developing signs of secondary bacterial infections [17].

The aim of our study was to determine the prevalence of antibiotic treatment in hospitalized adults with confirmed viral RTIs. In addition, we also investigated factors influencing the antibiotic decision-making in this large cohort of viral RTI patients from a regional general hospital.

## 2. Results

### 2.1. Infection Events (IE)

The study population consisted of patients with viral RTIs hospitalized 1222 times in total during the inclusion period (Figure 1).

A total of 12 cases of hospital-acquired infections were excluded, as well as 33 infection events (IE) with respiratory virus only detected in the readmission event. From the remaining 1177 IEs, 226 patients who received antibiotics due to bacterial infection were excluded.

During 2015–2018, 932 patients were hospitalized on 951 occasions due to viral RTIs. Of these 951 IEs, 720 received antibiotic treatment with no sign of bacterial infection and 231 did not receive antibiotic treatment during hospitalization.

Of the total IEs, 378 (39.7%) were hospitalized with influenza A, 231 (24.3%) with influenza B, 131 (13.8%) with respiratory syncytial virus (RSV), 122 (12.8%) with human metapneumovirus (hMPV), 81 (8.5%) with parainfluenza virus and 8 patients (0.8%) with adenovirus. In the 951 infection events included, no significant difference was found concerning gender when comparing patients who received antibiotic treatment to patients who did not receive such treatment during hospitalization (Table 1). The median age in the cohort was 73 years (range 18–103 years), with significant higher median in the patients treated with antibiotics (median 73 years, range 18–103) when compared to patients not treated (median 72 years, range 18–97), *p* = 0.037 (Wilcoxon rank sum test). The distribution over age groups was 3.6% in 18–29 years, 3.3% in 30–39 years, 5.8% in 40–49 years, 9.5% in 50–59 years, 17.4% in 60–69 years, 28.9% in 70–79 years, 25.4% in 80–89 years and finally 6.2% in the age group of ≥ 90 years.

In the group who received antibiotic treatment, the median Charlson score was 2 (range 0–9), while the median Charlson score in patients who did not receive such treatment was significantly lower with a median of 1 (range 0–8), *p* = 0.037 (Wilcoxon rank sum test) (Table 1). Concerning comorbidities, 34.7% suffered from chronic obstructive pulmonary disease (COPD)*,* and 15.2% and 14.0% suffered from prior myocardial infarction and congestive heart failure, respectively. Solid malignant tumors had been diagnosed in 9.9% of the cases and 1.9% suffered from hematopoietic malignancies. In addition, 9.3% and 1.6% of the patients had renal failure and liver failure, respectively, and 17.9% of the patients had diabetes. Furthermore, 209 cases (22.0%) were categorized as immunocompromised, either due to treatment or underlying condition. In the cohort, 2.3% of the patients were organ transplanted. Patients with COPD, immunosuppression and who were organ transplanted showed a significant difference with an overrepresentation in the antibiotic-treated group, with *p* < 0.001, *p* = 0.049 and *p* = 0.029, respectively (Pearson’s chi-square test).

The proportion with National Early Warning Score (NEWS-score) ≥ 5 points was significantly higher in the patients receiving antibiotics compared to those not treated (OR 4.12, 95% CI 2.70–6.28, *p* < 0.001, logistic regression analysis).

Both C-reactive protein (CRP) level >60 mg/L and white blood cell (WBC) count ≥ 11.1 10^9^/L were observed more often in the antibiotic-treated group than in the group of patients not treated (*p* < 0.001, Pearson’s chi square-test).

The length of the hospital stay (LOS) varied between 0 days (exceeding 5 h) and 27 days, with a median of 4 days. The median LOS was significantly longer in patients on antibiotic treatment compared with non-treated, 4 days vs. 2 days, respectively (*p* < 0.001, Wilcoxon rank sum test). Furthermore, in linear regression including age, gender and Charlson score, the antibiotic treatment had a significant coefficient of 2.24 (95% CI 1.70–2.77, *p* < 0.001), as antibiotic treatment showed an increase in LOS with on average 2.24 days.

Figure 2 shows a Kaplan–Meier plot of LOS in the group of patients treated with antibiotic treatment and in the patients who did not receive such treatment.

A total of 30 patients (3.2%) died during the hospital stay, and a further 23 patients (2.5%) died during the 30 days following discharge. The median age of the deceased was 81 years (range 57–103 years), 29 females (54.7%). Of these, two patients suffered from metastatic cancer. The median Charlson score was found to be 3 (range 0–8), significantly higher than surviving patients (median 2 (range 0–8), *p* < 0.001, Wilcoxon rank sum test). Antibiotic treatment was given to 83% of the deceased patients, 18 patients (34.0%) needed Intensive Care Unit (ICU) treatment during their hospital stay, 17 (32.1%) needed non-invasive ventilation and 3 patients (5.7%) required mechanical ventilation. Out of the deceased patients, 22 (41.5%) suffered from influenza A, 12 from influenza B (22.6%), seven from RSV (13.2%), eight from hMPV (15.1%) and four from parainfluenza (7.5%). No significant difference linked to antibiotic treatment status was observed in mortality during the hospital stay (*p* = 0.064, Pearson’s chi-square test) or 30 days after discharge (*p* = 0.881, Pearson’s chi-square test). In addition, antibiotic treatment was not associated with higher mortality, when adjusted for age, gender and Charlson score (OR 1.50, 95% CI 0.71–3.18, *p* = 0.289, logistic regression analysis).

### 2.2. Respiratory Virus Findings

Virological investigation of respiratory samples detected influenza A in 39.7% of cases, influenza B in 24.3%, RSV in 13.8%, hMPV in 12.8%, parainfluenza in 8.5% and only 0.8% adenovirus of the events (Table 1).

The time to results, calculated from the sampling time to the final laboratory report on the respiratory virus panel, was a median of 1 day (ranging from 0.3 days up to 4 days). The analysis was not performed during weekends and holidays.

For 21 samples (2.2%), the turn-around time ranged from 5 to 17 days, with median of 7 days, due to delay in transport to the laboratory or technical problems. Of these 21 cases, 18 (86.0%) received antibiotic treatment; among these 18 cases, 5 had already discontinued antibiotics before the test result was returned. In four other cases, the antibacterial treatment was discontinued the same day or the day following the test result.

### 2.3. Antimicrobial Drug Treatment

#### 2.3.1. Antibiotic Treatment during Hospital Stay

During the hospital stay, 720 (75.7%) of the 951 cases were prescribed antibiotics. Antibiotic prescription ranged from 71.0% in the influenza B-positive group to 83.2% in the RSV group.

In total, 15.0% of patients received antiviral treatment with oseltamivir, with no significant difference between the patients in the antibiotic-treated group and those who did not receive antibiotic treatment.

#### 2.3.2. Antibiotic Types Prescribed

Table 2 shows antibiotic prescription rates according to antibiotic types in the total antibiotic-treated cohort.

Of the 720 cases treated with antibiotics, 383 (53.2%) received only one antibiotic agent. Of the 337 cases receiving two or more agents, 32.1% were treated with two agents, 11.5% with three agents, 1.8% with four agents, 1.3% with five agents and 0.1% with six agents.

It is apparent from this table that most cases were prescribed with various types of penicillins, the most prevalent types ordained were beta-lactamase sensitive penicillins (46.7%) and extended-spectrum penicillins (42.8%), such as ampicillin and amoxicillin. Cephalosporins were prescribed in 23.1% of the IEs, and in 114 cases (15.8%), cephalosporins were prescribed as first-line treatment. Gentamicin was prescribed in 18.8%, but only in combination with penicillin.

Of the 720 patients who received antibiotic treatment, 291 patients (40.4%) received an antibiotic agent from the WHO Watch category [18]. The patients receiving Watch treatment had a similar age compared to those receiving Access agents, but had significantly higher median CRP than patients in the Access category (72 mg/L vs. 54 mg/L, *p* = 0.003, Wilcoxon rank sum test), and more comorbidities with a higher Charlson score (*p* = 0.028, Wilcoxon rank sum test). They were more frequently in need of ICU treatment (*p* = 0.003, Chi-square test), and had a higher in-hospital mortality rate, 18 vs. 9 (*p* = 0.005, Chi-square test).

Considering length of treatment (LOT), adjusted for age, gender and comorbidity, receiving a Watch agent prolonged the LOT with 1.4 days compared to an Access agent (coefficient 1.41, 95% CI 0.83–1.99, *p* < 0.001). In addition, patients who received a Watch agent had on average a 1.67 days longer hospital stay compared with patients receiving Access agents, adjusted for age, gender and comorbidity (95% CI 1.11–2.23, *p* < 0.001).

#### 2.3.3. Length and Days of Antibiotic Treatment

The length of antibiotic treatment (LOT, days from the date of the first dose to the date of the last dose delivered) ranged from 0.5 to 36 days with a median duration of 7 days.

In the total cohort, 212 (30.5%) had a LOT of 0–5 days, while in 484 (69.5%) of the infection events treatment for six days or longer was prescribed.

Days of therapy (DOT, the total number of days a patient receives an antibiotic during a treatment course) ranged from 0.5 to 40 days, with median 8 days. Data on LOT and DOT were missing in 24 cases (3.3%).

#### 2.3.4. Discontinuation of Antibiotic Treatment

In the minority of patients, 107 (15.8%), treatment was discontinued quickly (within 1 day) after confirmation of a viral RTI. In 439 cases (64.7%), antibiotic treatment was terminated after five days or more after the confirmation of a viral RTI, ranging up to 35 days. In 18 cases, the treatment was stopped before the confirmation of a viral RTI, and in 24 cases, the date for the end of treatment was not found in the patient records.

In the patient group hospitalized with influenza A, the antibiotic treatment was terminated within 1 day after viral RTI confirmation in 14.0% of the cases. For patients with influenza B and adenovirus, the antibiotic termination rate was found to be 12.8% and 12.5%, respectively, RSV 10.4%, parainfluenza 8.9% and for patients with hMPV 6.1%.

Compared to influenza A, the only virus found to be statistically different in termination rate was hMPV, with OR of 0.35 (95% CI 0.15–0.80, *p* = 0.013, logistic regression analysis).

In most of the patients treated with antibiotics, the treatment continued after discharge from the hospital (n = 449, 64.5%).

#### 2.3.5. Evaluation Score

As described in Section 4.3, we designed an evaluation score to investigate the relevance and indication for prescribing antibiotic treatment. Table 3 shows the distribution of scores for these five factors potentially influencing decision-making, showing a median score of 2 (range 0–5).

In the group treated with antibiotics (n = 720), the median evaluation score was two (0–5), and 65.8% of the patients scored ≥ 2 points. In the group of patients not treated with antibiotics, the median score was 1 (0–4), and only 24.7% of these patients scored ≥ 2 points. The difference in the evaluation score median was found statistically significant (*p* < 0.001, Wilcoxon rank sum test).

Using regression analysis adjusted for age and gender, we found a significantly higher use of antibiotic treatment among patients with an evaluation score of two or more, with OR of 5.80 (95% CI 4.13–8.13, *p* < 0.001).

Table 4 shows the results of logistic regression analysis of the five factors included in the evaluation score, and their association with antibiotic prescription during hospital stay. Two of the evaluation score factors were significantly more often found in patients prescribed antibiotics: signs of pneumonia in chest X-ray/CT (OR 3.66 (95% CI 2.40–5.58, *p* < 0.001) and CRP level >60 mg/L (OR 4.69 (95% CI 3.12–7.04, *p* < 0.001). Receiver Operating Characteristic (ROC) curve analysis of the evaluation score showed an Area Under the Curve (AUC) of 0.755.

Both immunocompromised patients and patients without such a risk factor had the same median evaluation score of two (range 0–5), with no significant difference (*p* = 0.150, Wilcoxon rank sum test). Further, we found that patients in need of a stay in the intensive care unit (ICU) had a significantly higher evaluation score median of 3 (range 0–5) compared to patients not in need of the ICU with a median evaluation score 2 (range 0–5)) (*p* < 0.001, Wilcoxon rank sum test), and only 4.4% of these received no antibiotic treatment.

Considering the length of stay (LOS), adjusted for age, gender and Charlson score, we found a coefficient of 0.90 (95% CI 0.71–1.08, *p* < 0.001, linear regression analysis) for the evaluation score, each additional point in the evaluation score leads to an average increase of 0.9 days longer hospital stay (LOS). When further adjusting for antibiotic treatment, each additional point increase in evaluation score added 0.7 days to total LOS.

## 3. Discussion

In Norway, the prevalence of antibiotic resistance is low from a global perspective. However, a national aim is still to reduce antibiotic prescriptions and especially the use of broad-spectrum agents [19,20]. This study shows that most patients (75.7%) hospitalized due to viral RTIs received antibiotic medication, and that the majority continued their course for five days or longer even after the laboratory confirmation of a viral etiology. Additionally, discontinuation within one day after diagnosis occurred in only 15.8% of the patients treated with antibiotics without microbiological evidence of bacterial infection. Thus, the termination of antibiotic therapy after detection of a viral RTI was often delayed, and in most cases, an entire treatment course was completed. In addition, we found that prescription of antibiotics leads to a longer hospital stay and mortality was not affected by antibiotic treatment status.

The national antibiotic guidelines for the hospital sector in Norway are based on scientific evidence, treatment tradition and national antibiotic-resistance patterns. Empiric treatment when suspecting community-acquired pneumonia (CAP) is narrow-spectrum antibiotics, i.e., benzylpenicillin. For severe CAP and non-complicated sepsis, benzylpenicillin or ampicillin in combination with gentamicin is recommended [20]. Our study findings are in accordance with the national guidelines, as the choice of empiric treatment showed that the most used antibiotic type was beta-lactamase-sensitive penicillins or extended-spectrum penicillins such as ampicillin. Cephalosporins were prescribed as the first-line antibiotic in 16% of the cases and according to the national guidelines, cephalosporins should only be prescribed as a second-choice treatment, or in case of very severe CAP. Adherence to the antibiotic prescription guidelines is strongly advised but may have been disregarded in some of these cases due to high CRP values, NEWS-score or high severity.

In 2017, WHO developed the AWaRe classification of antibiotics as a tool to support antibiotic stewardship efforts at local, national and global levels [18]. In AWaRe, antibiotics are classified into three groups, Access, Watch and Reserve, according to their impact on antimicrobial resistance, to emphasize the importance of their appropriate use [18]. Our study found that 40.4% of the patients treated with antibiotics were prescribed an agent belonging to the Watch category, i.e., antibiotic classes with higher resistance potential, and thus should be prioritized for stewardship and monitoring. These patients had a higher CRP level and Charlson score, were more often in need of ICU treatment and had higher in-hospital mortality. In addition, both length of treatment and length of stay at hospital were prolonged. None of the patients were prescribed antibiotics in the Reserve category, which is an antibiotic category that should be reserved for the treatment of infections due to multi-drug resistant organisms. Our results show room for improvement regarding antimicrobial stewardship in our region with respect to adherence to local, national and global action plans for antibiotic consumption.

According to the national antibiotic guidelines for primary care in Norway [21], and also the British guidelines for CAP [22], it is recommended to initiate antibiotic treatment when suspecting CAP. The British guidelines stress that empirical broad-spectrum antibiotics are initially recommended only in patients with high-severity CAP. It is also pointed out that a senior clinician should review the diagnosis of CAP and the decision to start antibiotics at the earliest opportunity, and that there should be no barrier to discontinuing antibiotics if not indicated.

On the other hand, the detection of a viral respiratory infection does not exclude a bacterial respiratory infection, as secondary bacterial infection can arise during the course of illness [23,24] and arise in up to 40% of viral respiratory tract infections requiring hospitalization [25]. In a recently published systematic review of CAP, respiratory viruses were reported to be found in about 30% of CAP in adults, with substantial rates of viral/bacterial coinfection [26]. The authors concluded that the identification of a virus by PCR does not, by itself, allow for discontinuation of antibiotic therapy. Another study, from Norway and investigating the microbiological etiology of CAP, found bacterial etiology in 47%, viral etiology in 34% and viral bacterial coinfections in 31% of the patients [27]. Access to rapid molecular diagnostics involving a panel consisting of both clinically relevant bacterial and viral agents is therefore important, especially because the standard culture methods for growth of bacteria is time consuming and may further extend the time to antibiotic discontinuation.

We also found that patients in need of treatment in the ICU department during their hospital stay had significantly higher evaluation scores compared to patients not needing ICU admission. This may reflect that the variables included in the evaluation score is a suitable predictor for severity of the disease.

In our study, 362 patients (37%) had radiological findings (chest CT or chest X-ray) characteristic of pneumonia, and regression analysis showed a highly significant OR of initiation of antibiotic treatment, in addition to CRP level > 60 mg/L. Radiographic results may be of uncertain value when differentiating between viral and bacterial etiology. Studies have shown that viral pneumonia also presents with similar radiological findings as certain types of bacterial pneumonia [28,29]. Even though some characteristics may help to differentiate particular viral RTIs, there is considerable overlap in the imaging appearance of viral and bacterial respiratory infections [30].

In addition, in our study we found that in only 27 cases (2.8%) were neither chest X-ray nor CT scans performed during the hospital stay. A recently published multicenter study aimed to investigate the influence of X-ray results on antibiotic prescription in the pediatric population [31]. They concluded that making the decision of performing a chest X-ray was independently associated with a higher rate of antibiotic prescription regardless of the radiological findings, showing the inferior role of chest X-ray results in treatment decisions.

The CRP level is another deciding factor, which in many cases may be of uncertain value in distinguishing between bacterial and viral etiology. The sensitivity and specificity of the CRP test in bacterial pneumonia is relatively low, and CRP values above the normal range can be due to other illnesses [32]. Additionally, there is sparse and diverging information in the literature on the level of CRP for the diagnosis of bacterial pneumonia in adults; some studies suggest 60 mg/L as cut-off and others use 100 mg/L [12,33]. In addition to CRP, the host-response marker procalcitonin may be valuable in the etiology differentiation, and appears to be the earliest marker to appear during infection. Clinical trials have shown that the presence of an elevated level (>0.25–0.5 μg/L) can be used to identify patients requiring treatment for pneumonia [34]. Procalcitonin was not available as a routine test at Østfold Hospital during the study period. The test was later added as a useful antibiotic stewardship tool, assisting clinicians in the difficult considerations of initiating, escalating or de-escalating antibiotic therapy [35,36].

In many situations, antibiotic initiation happens early in the course of infection based solely on a clinical evaluation, which in most cases cannot distinguish between bacterial and viral RTIs [13,14,37]. Clinical observations such as respiratory rate, heart rate and oxygen saturation are potential factors that may help the clinician in evaluating the severity of the infection, but not the etiology. In a recently published study on hospitalized patients with an exacerbation of COPD and confirmed influenza or RSV infection, the authors concluded that prescription of antibiotics in COPD patients is a common practice despite a proven viral infection on admission [38]. Our study supports these findings, as approximately 86% of the patients suffering from COPD received antibiotic treatment. The authors also suggested that a 34.2% reduction in antibiotic prescription in this cohort was feasible [38]. Another study, investigating a cohort of 196 hospitalized adults with viral RTIs, showed that of the 131 patients who were administered antibiotics, 125 continued to receive the treatment after receiving a viral diagnosis [39]. In this study, an abnormal chest radiograph was independently associated with continued antibiotic use. A study from Israel, investigating hospitalized adults with influenza or RSV infection, found that antibiotic therapy was administered to 77% of the patients, despite that only 32% had a concomitant bacterial infection [40]. In the patients without bacterial infection, the antibiotic treatment was stopped when viral RTI results were reported in 37% of the cases [40], which is a considerably higher percentage than found in our study.

A study from Spain concluded that not only did RSV patients more frequently receive antibiotic treatment when compared to influenza patents, the antibiotic withdrawal at the time of diagnosis was lower among these patients [41]. In our study, we found that the discontinuation rate in the antibiotic-treated RSV patients, with no sign of bacterial infection, was 10%, as opposed to 14% of the influenza A patients, but the difference was not statistically significant. A study from Østfold Hospital Trust (2018–2019) explored factors affecting hospital physicians’ antibiotic prescribing practices using semi-structured interviews. The study revealed that, among other factors, suboptimal microbiological testing affected antibiotic choice [42]. The authors concluded that to limit the broad-spectrum antibiotic use, improving microbiology testing and the routines for consultations with infectious disease specialists would be beneficial.

The British Thoracic Society recommends seven days of appropriate antibiotic therapy for patients with low- or moderate-severity community-acquired pneumonia [22]. The IDSA/ADS (Infectious Diseases Society of America/American Thoracic Society) guidelines state that for adults with CAP, a minimum of at least five days of antibiotic therapy is recommended [43]. The patients in our study had a median LOT of seven days, ranging up to 36 days, and 69.5% received antibiotic therapy for 6 days or longer. This may imply both excessive use and hesitation to discontinue treatment. Clearly, this calls for more focus on antibiotic stewardship to reduce the length of therapy, which in addition, could shorten hospital stays.

The median turn-around time for laboratory results on respiratory viruses was only one working day in this study. A short time-lag to viral diagnosis did not lead to antibiotic de-escalation for the majority of our patients, which may be due to the more time-consuming bacterial diagnostics, and clinicians’ hesitancy to discontinue treatment prior to culture results. Access to rapid laboratory testing of a suitable panel of both viral and bacterial respiratory agents, according to the epidemiological situation, is important. Fast and pertinent microbiological testing is crucial to help clinicians make the right treatment decision for the patients’ benefit [44]. Rapid diagnostic testing is also needed to help decide on prompt isolation precautions to prevent nosocomial infections and can minimize unnecessary initiation of antibiotics and reduce the duration of antibiotic treatment, ultimately preventing the development of antibiotic resistance.

Our study has some limitations. This study is a single-center study, so generalization of our findings must be carefully considered. We did not use any questionnaires for clinicians to gather information on reasons for continuing antibiotics after diagnosis of a viral RTI. Observational studies have a potential for bias due to using data collected retrospectively. As this study was designed to investigate patients presenting with respiratory symptoms, the antibiotic agents mainly used to treat urinary tract infections (sulfamethoxazole and trimethoprim, mecillinam/pivmecillinam and nitrofurantoin) were excluded from analyses. Recognizing that some antibiotic agents, such as cephalosporins, are used to treat cases with suspicion of concomitant RTIs and UTIs, we may have overrated the number of patients given antibiotics to treat RTIs.

In addition, the study includes a few patients who may have been admitted with neutropenic fever. In these patients, empiric antibiotic treatment is often initiated. Even in the absence of bacterial growth, a broad-spectrum antibiotic treatment will, as a rule, be given in its full course. In addition, empiric antibiotic treatment is often necessary at the onset of fever in patients with underlying illnesses or drug-induced immunosuppression.

The possibility of a bacterial infection despite negative cultures cannot be excluded. In addition, the lack of microbiological evidence of a bacterial infection may reflect insufficient microbiological sampling.

Procalcitonin could have been an important factor in the decision-making regarding antibiotic initiation, but unfortunately this test was not available as a routine test in the hospital in the study period. However, the data on microbiological results and antibiotic treatment gave the basis for a comprehensive evaluation of a cohort of patients admitted to a large hospital serving the entire population in the region.

## 4. Materials and Methods

### 4.1. Study Population

Østfold Hospital Trust is a 500-bed, secondary acute care hospital with a catchment area of approximately 320,000 inhabitants. We conducted a retrospective observational single-center study, and included patients admitted to the hospital with viral RTIs during the years 2015–2018 [45]. The patients were identified from the laboratory database by the finding of a virus in the multiplex polymerase chain reaction Seegene Allplex™ Respiratory Panel 1 and 2 assay, which detects influenza A, influenza B, respiratory syncytial virus, human metapneumovirus, parainfluenza virus 1–4, adenovirus and enterovirus. Infection events (IE) were defined as patients presenting at hospital admission with the most common symptoms of respiratory virus infections (e.g., cough, fever, dyspnea, fatigue, malaise, running nose), in a time period of 14 days preceding or following a positive virus finding. Readmission with a new virus finding was categorized as a new IE. Readmissions with the same virus finding were not included, as these were considered to belong to the same IE.

We excluded cases with a finding of two viral agents, and in addition cases of enterovirus infections, because of the uncertainty of the clinical relevance of this virus agent in respiratory samples [46].

Hospitalized patients who presented symptoms of respiratory tract infection >48 h from admission were considered hospital-acquired infections (HAI) and were excluded, as well as the IEs with respiratory virus only detected in the readmission event. A bacterial infection was diagnosed if microbiological examination revealed growth of relevant pathogenic microbes in respiratory specimen or blood culture, or by legionella/pneumococcus urine antigen or Mycoplasma/Chlamydia pneumonia PCR positive tests. These patients were excluded, leaving the final study population of hospitalized patients in two groups; antibiotic treated with no sign of bacterial infection and patients who did not receive antibiotic treatment during hospitalization.

Concomitant medical conditions were registered and standardized using the Charlson comorbidity index, an assessment tool designed specifically to predict long-term mortality. The Charlson comorbidity index includes 19 medical comorbid conditions, with a weighted score of either 1, 2, 3 or 6 assigned to each condition, based on the adjusted relative risk associated with each comorbidity to predict 1-year mortality [47,48,49].

Information on immunosuppression was registered according to hospital records. A patient was categorized as immunosuppressed if treated with immunosuppressive medicines (such as methotrexate or rituximab), organ transplanted, suffering from disorders of the immune system, received cancer chemotherapy 1 month or less prior to admission or treated with systemic corticosteroids 14 days or less prior to admission.

NEWS (National Early Warning Score) is a tool to quickly detect changes in the patient’s vital signs and clinical condition to prevent deterioration. Therefore, we evaluated the disease severity by reviewing the first NEWS value registered at admission, with a special analysis of the number of cases with ≥5 points, which is considered the critical threshold value for urgent medical response [50].

We retrospectively scored the patients by using CRB-65, which is a score system tool to estimate the severity and risk of death of community-acquired pneumonia [51].

### 4.2. Laboratory Findings and Data on Antibiotic Prescription

Respiratory samples were analyzed for respiratory viruses by using Seegene Allplex Respiratory Panel 1 and 2 assays (Seegene Inc., Seoul, Republic of Korea) as previously reported [45,46]. The time from reception of specimens for respiratory virus testing to laboratory reporting was calculated, including weekends and holidays. As a routine, the Seegene Allplex assay is performed every working day, and the total analyzing time including extraction is approximately 3.5 h. Laboratory findings were drawn from the laboratory information system (LIS) LVMS (Lab Vantage Medical Suite).

Data on the administration of antimicrobial drugs during hospitalization were collected from the electronic hospital records and analyzed using a purpose-built structured database. If the date for the end of antibiotic treatment was not found in the hospital record, these patients were excluded in the calculation of DOT and LOT. Antibiotic agents specifically aimed for urinary tract infections (J01E E01 sulfamethoxazole and trimethoprim, J01C A08 pivmecillinam, J01C A11 mecillinam, J01X E01 nitrofurantoin), and treatment of *Clostridioides difficile* infection were excluded. In addition, we excluded cloxacillin/dicloxacilllin treatment when administered together with a microbiological finding of staphylococcus in soft tissue, blood culture or urine.

### 4.3. Antibiotic Prescription Factors—Evaluation Score

Due to the lack of a validated tool to evaluate antibiotic prescription retrospectively, an evaluation score was designed as a helping tool to better understand the decision-making in this cohort. We included clinical-, biochemical- and radiological factors that clinicians seem to emphasize when deciding whether to prescribe antibiotics, to investigate the relevance and indication of the antibiotic treatment. The infection events were scored as follows:Chest X-ray/Chest CT compatible with or susceptible of pneumonia (1 point);CRP level > 60 mg/L at admission (1 point);Oxygen saturation < 90% at admission (1 point);Respiratory rate > 20/min at admission (1 point);Heart rate > 100 at admission (1 point).

The diagnosis of pneumonia by chest X-ray or CT was made by a medical specialist in radiography at the time of the examination during the hospital stay.

### 4.4. Ethics Permission

Ethical approval was obtained from the Regional Committee for Medical and Health Research Ethics (REK ref. 2017/1917 A) and the hospital Privacy Appeal Board (public 17/05444).

### 4.5. Statistical Analyses

For statistical analyses, we used SPSS (IBM SPSS Statistics for Windows, version 25.0. Armonk, NY, USA: IBM Corp.).

Categorical variables are presented as frequencies and percentages; continuous variables are described as mean, median, standard deviation and range as appropriate.

Statistical analysis was performed on categorical variables by using Pearson’s chi-square test, and Fisher’s exact test for variables with expected count n ≤ 5, for continuous variables Wilcoxon rank sum test was performed. We adjusted for patient age and gender in all regression analyses. Length of stay analysis is presented as Kaplan–Meier plot.

ROC curve analysis was performed on the evaluation-score.

Normality examination was performed with Shapiro–Wilk test.

*p*-values less than 0.05 were considered statistically significant.

## 5. Conclusions

Our study revealed that antimicrobial stewardship and rational antibiotic prescription could be improved even in a country with relatively low antibiotic usage. Doctors were reluctant to stop antibiotic treatment in cases with viral infection, even after receiving negative bacterial culture results, as most cases continued with a full treatment course.

In addition, length of stay was significantly higher in the group of patients on antibiotic treatment compared to those not treated, showing that patients may benefit from more stringent antibiotic stewardship by a shorter hospital stay. Additionally, the fact that mortality was similar in both groups supports that not giving antibiotics in these cases is safe and reassuring for physicians.

Together, these results provide important insights into medical decisions governing the antibiotic administration and can be useful for improvement in the management of viral RTIs.

Access to relevant microbiological tests and rapid communication of results are crucial for reducing the antibiotic prescription. Development of better tools for antibiotic stewardship needs to be further investigated.

## Figures and Tables

**Figure 1 antibiotics-12-00788-f001:**
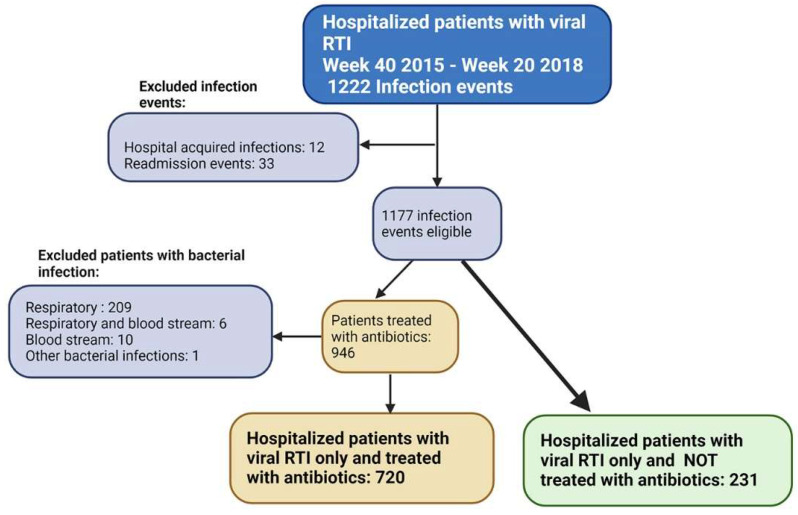
Flowchart showing the selection of the study population.

**Figure 2 antibiotics-12-00788-f002:**
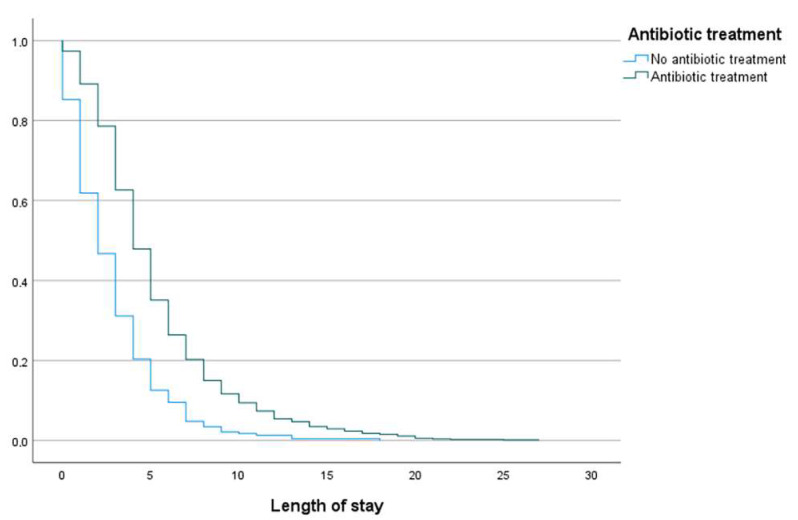
Kaplan-Meier analysis on length of stay comparing the groups of patients according to antibiotic treatment.

**Table 1 antibiotics-12-00788-t001:** Demographic data, medical status at admission and clinical outcome in 951 hospitalizations due to viral RTIs according to antibiotic treatment status.

	Total N = 951	Antibiotic Treated n = 720	Not Antibiotic Treated n = 231	*p*-Value *
Female, n (%) Male, n (%)	506 (53.2) 445 (46.8)	385 (53.5) 335 (46.5)	121 (52.4) 110 (47.6)	0.772
Age, years median (range)	73 (18–103)	73 (18–103)	72 (18–97)	0.037
Age ≥ 65 years (%)	662 (69.6)	518 (71.9)	144 (62.3)	0.006
Charlson Score, median (range)	2 (0–9)	2 (0–9)	1 (0–8)	0.037
Chronic obstructive pulmonary disease, n (%)	330 (34.7)	283 (39.3)	47 (20.3)	<0.001
Immunosuppressed, n (%)	209 (22)	169 (23.5)	40 (17.3)	0.049
CRB65 category, median (range) Mean (±SD)	2 (1–3) 1.79 (±0.50)	2 (1–3) 1.83 (±0.50)	2 (1–3) 1.67 (±0.49)	<0.001
Patients with temperature ≥ 38 C, n (%)	497 (52.3)	408 (56.7)	89 (38.5)	<0.001
Oxygen saturation at admission, median (range)	94 (57–100) Nd = 1	94 (57–100)	96 (79–100) Nd = 1	<0.001
Patients with oxygen saturation < 90%, n (%)	167 (17.6) Nd = 1	149 (20.7)	18 (7.8) Nd = 1	<0.001
CRP level at admission, median (range)	50 (1–589) Nd = 1	61.5 (1–589)	24 (1–307) Nd = 1	<0.001
Patients with CRP level > 60 mg/L, n (%)	401 (42.2) Nd = 1	362 (50.3)	39 (16.9) Nd = 1	<0.001
Patients with WBC level ≥ 11.1 10^9^/L, n (%)	222 (23.3) Nd = 3	199 (27.6) Nd = 1	23 (10.0) Nd = 2	<0.001
NEWS score, median (range)	4 (0–13) Nd = 90	4 (0–13) Nd = 44	2 (0–10) Nd = 46	<0.001
NEWS ≥ 5 points, n (%)	331 (34.8) Nd = 90	301 (41.8) Nd = 44	30 (13.0) Nd = 46	<0.001
Non-invasive ventilation (CPAP, BiPAP), n (%)	100 (10.5)	95 (13.2)	5 (2.2)	<0.001
Mechanical ventilation, n (%)	5 (0.5)	5 (0.7)	0 (0)	0.344
Admitted to ICU, n (%)	112 (11.7)	107 (14.9)	5 (2.2)	<0.001
Antiviral treatment (oseltamivir), n (%)	143 (15.0)	105 (14.6)	38 (16.5)	0.490
Median length of stay, days (range)	4 (0–27)	4 (0–27)	2 (0–18)	<0.001
Length of stay ≥ 5 days	394 (41.4)	346 (48.1)	48 (20.8)	<0.001
Death during hospital stay, n (%)	30 (3.2)	27 (3.8)	3 (1.3)	0.064
All cause mortality 30 days after discharge, n (%)	23 (2.5)	17 (2.5)	6 (2.6)	0.881

* *p*-value: For categorical variables: Pearson’s Chi-square test, Fisher’s exact test for variables with expected count n ≤ 5. For continuous variables: Wilcoxon rank sum test; Abbreviations: SD = Standard deviation; WBC = White blood cell count; CRP = C-Reactive Protein; NEWS = National Early Warning Score; CPAP = Continuous Positive Airway Pressure; BiPAP = Bilevel Positive Airway Pressure; ICU = Intensive Care Unit.

**Table 2 antibiotics-12-00788-t002:** Antibiotic prescription rates in 720 patients according to categories of antibacterial drugs and the AWaRe classification.

	N = 1200	%	AWaRe *
J 01C E Beta-lactamase Sensitive Penicillins ^a^	336	46.7	Access
J 01C A Extended spectrum Penicillins ^b^	308	42.8	Access
J01D B First generation Cephalosporins ^c^	2	0.3	Access
J01D C/D Second- and third generation cephalosporins ^d^	164	22.8	Watch
J01G B03 Gentamicin	135	18.8	Access
J01F A Macrolides ^e^	117	16.3	Watch
J01 A A02 Doxycycline	41	5.7	Access
J01C R05 Piperacillin and beta-lactamase inhibitor ^f^	24	3.3	Watch
J01C R02 Amoxicillin and beta-lactamase inhibitor ^g^	3	0.4	Access
J01M A02 Ciprofloxacin	26	3.6	Watch
J01X D Metronidazole	10	1.4	Access
J01F F Clindamycin	16	2.2	Access
J01D H Carbapenems ^h^	13	1.8	Watch
J01X A01 Vancomycin	4	0.6	Watch
J01C F Beta-lactamase Resistant Penicillins ^i^	1	0.1	Access

* WHO Access, Watch, Reserve (AWaRe) classification of antibiotics for evaluation and monitoring of use [18]. ^a^ Benzylpenicillin, Phenoxymethylpenicillin; ^b^ Ampicillin, Amoxicillin, Imacillin; ^c^ Cephalothin; ^d^ Cefotaxime, Ceftriaxone, Cefuroxime, Ceftazidime; ^e^ Erythromycin, Azithromycin, Clarithromycin; ^f^ Piperacillin/Tazobactam; ^g^ Amoxicillin/Clavulanic acid; ^h^ Meropenem, Ertapenem; ^i^ Cloxacillin, Dicloxacillin.

**Table 3 antibiotics-12-00788-t003:** Five factors potentially influencing initiation of antibiotic treatment (evaluation score 0–5 points) in 951 infection events.

	Total N = 951	Antibiotic Treated n = 720	Not Antibiotic Treated n = 231	*p*-Value *
Chest X-ray/Chest CT compatible or susceptible of pneumonia, n (%)	362 (37.0) Nd = 27	330 (45.8) Nd = 8	32 (13.9) Nd = 19	<0.001
CRP level > 60 mg/L, n (%)	401 (42.2) Nd = 1	362 (50.3)	39 (16.9) Nd = 1	<0.001
Oxygen saturation < 90%, n (%)	167 (17.6) Nd = 1	149 (20.7)	18 (7.8) Nd = 1	<0.001
Heart rate > 100/min, n (%)	295 (31.0)	242 (33.6)	53 (22.9)	0.002
Respiratory rate > 20/min, n (%)	474 (49.8)	392 (54.4)	82 (35.5)	<0.001
Number of evaluation score points				
0 points	148 (15.6)	67 (9.3)	81 (35.1)	
1 point	272 (28.6)	179 (24.9)	93 (40.3)	
2 points	267 (28.1)	223 (31.0)	44 (19.0)	
3 points	178 (18.7)	169 (23.5)	9 (3.9)	
4 points	71 (7.5)	67 (9.3)	4 (1.7)	
5 points	15 (1.6)	15 (2.1)	0 (0)	
Evaluation score median (range)	2 (0–5)	2 (0–5)	1 (0–4)	<0.001
Evaluation score ≥ 2 points	531 (55.8)	474 (65.8)	57 (24.7)	<0.001

* *p*-value: For categorical variables: Pearson’s chi-square test; For continuous variables: Wilcoxon rank sum test; CRP = C-Reactive Protein; Nd = No data.

**Table 4 antibiotics-12-00788-t004:** Results of potentially factors influencing initiation of antibiotic treatment (outcome) by logistic regression.

	OR	95% CI	*p*-Value
Age	1.012	1.002–1.022	0.022
Gender female	0.948	0.674–1.332	0.757
Chest X-ray/chest CT compatible or susceptible of pneumonia	3.658	2.399–5.580	<0.001
CRP level > 60 mg/L	4.686	3.118–7.043	<0.001
Oxygen saturation < 90%	2.107	1.210–3.669	0.008
Heart rate > 100/min	1.537	1.031–2.290	0.035
Respiratory rate > 20/min	1.762	1.236–2.512	0.002
Constant	0.400		0.018

Hosmer and Lemeshow test: *p* = 0.999.

## Data Availability

The data presented in this study are available on request from the corresponding author.

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
