# Peer review of "Antibiotic Consumption in a Cohort of Hospitalized Adults with Viral Respiratory Tract Infection"

_antibiotics, 2023, doi:10.3390/antibiotics12040788_

Round 1

Reviewer 1 Report

The authors tried to illustrate that even in patients admitted with respiratory viral infections in a Norweigein hospital (where antibiotic use is in general low), antibiotic use is high 75% of cases, and there is still potential for improved antibiotic use. 

There are however a few important points that the authors have to clarify.

1. How was appropriateness of antibiotic use determined? How was bacterial infection determined? (? Microbiological proven?) what abt the possibility of possible bacterial infection but culture negative. As mentioned, procal is not available, so how was the arbitration performed? These questions will affect the author’s analysis and conclusions. 

2. The title is misleading. “Antibiotic decision-making in a cohort of hospitalised adults with viral RTI”

- IT reads as if the authors were proposing a set of guidance to recommend antibiotic use, when in reality, they are reviewing factors which might drive prescribers to start antibiotics, and some of the reasons may or may not be valid.

- Although the intent was very nicely pointed out in lines 25-26, this message was not consistent throughout the text. E.g. lines 31-32 - what the authors did was to review clinical, laboratory and radiological factors that might drive antibiotic use, but to design a decision score implies making a recommendation for antibiotic use and it is 2 different things. 

3. The crux of the matter is - was when and how was the diagnosis of viral RTI made? And was antibiotic discontinued after the diagnosis was made ?This information is hard to pick up on first read, and has to be mined. So the results of the study have to be interpreted with caution; and application for clinical practice not immediately obvious.

4. It would be good to recommend interventions for stewardship based on your observational findings, and this is hard to appreciate in the paper. This has to be discussed.

Clarifications:

- How did the authors define absence of bacterial infection? (Line 116?) - this would be crucial, esp at the start, because it would affect how readers interpret your findings and recommendations. From the text, it appears that the population described is large and heterogenous, so could empiric bacterial treatment be necessary at the start? The key question is how do we use antibiotics safely and what approach should be adopt? A de-escalation or escalation approach (should antibiotics be started empirically at onset of illness, or when there is more definitive evidence for a bacterial infection)

- Why were various comparisons made between different viruses? What were the authors trying to illustrate? If this comparison is made, then the rationale has to be better discussed. 

- Results:

(a) —- CRP and WBC thresholds were described - cutoffs were described, presumably those were associated more with bacterial infection. But can the authors explain why those cutoffs were selected, with references where available?

(B) Antimicrobial use

- Why were non-respiratory antibiotics prescribed? Did the patients have a secondary bacterial infection at a distant site?, again, why did we compare antibiotic use across different respiratory viruses? Why was penicillin combined with gentamicin? Is this practised in Norway? The authors mentioned that penicillin in combination with gentamicin is used for uncomplicated sepsis, does this also apply to respiratory tract infections?

(C) from the length of antibiotic use discussion - it is almost apparent that antibiotic use is excessive… and this is something the team could work or intervene on. PLease also work on the messaging throughout the text (e.g. lines 269 - 270), lines 274 - 276.

Language:

- Could be improved throughout text to convey the right message and meaning. E.g.

—- Lines 41-42: I would replace with “… there is still a role for antimicrobial stewardship to further improve antibiotic use in patients admitted for viral RTIs in a country with relatively low antibiotic consumption”

—- Lines 52: increasing global health threat in place of global increasing health threat.

—- Introduction paragraph 2 (lines 62 - 68: very difficult to read and understand. Inappropriate antibiotic use is driven largely by non-adherence to guidelines. And i gather that the authors want to highlight causes of inappropriate antibiotic use in this paragraph, just make it succinct and clear.

—- Lines 154 - 159 - please rephrase to make it clearer. the gist is that those who received antibiotics inappropriately stayed longer. Just simplifying the text. 

—- LInes 201 - 203: I thought the authors wanted to highlight that in general the median time to a respiratory virus PCR results is only 1 day, and it can be longer over the weekend - just simplify the paragraph, and this would have bearing on stewardship interventions. This could be very elegantly discussed in discussion.

Organization of text:

—- Lines 85-91: Author mentioned abt viral RTI management (largely being supportive), then talks abt bacterial RTI treatment, then circle back to anti-viral management (please improve the flow).

Is empiric treatment for influenza practised in Norway or is it after viral PCRs are available? Perhaps the authors would like to offer some clarity?

- The stats was explained in the results section (lines 126). Could this be in methods?

- In the discussion section, the authors try to discuss a few points:

- CRP  / medical comorbities / procal use —-> in lines 408 - 421, 466 - 469: can this information be organised better? The discussion on CRP  / Procal should be grouped, and cutoffs to guide antibiotic use must be discussed. Also, are medical co-morbidities alone sufficient to drive antibiotic use? What abt after the diagnosis of a respiratory viral infection is made? All these nuances can be better discussed.

Abbreviations:

- Can the abbreviations be illustrated clearer in the text? IE is used in the results section but the abbreviation is explained in the later part of the text in the methods section. Difficult to follow. Is IE then infective endocarditis?

Reviewer 2 Report

Interesting topic. Reserach well done and very informative.

The manuscript touches on a very important topic - the use of antibiotics in viral respiratory infection.
The submitted manuscript is well written and the language is understandable. The obtained results are presented in an acceptable way in tables and figures that do not repeat the information already written. The methods used are well researched. The discussion of the obtained results is adequate and sufficient. The conclusion corresponds to the presented results. The cited references are up-to-date and well-chosen. In addition, introducing a score for decision making is an advantage of the paper. I recommend Accept.

Reviewer 3 Report

In introduction section, there is need to add data describing local data on antibiotics use particularly in hospitalized patients. It would be interesting if authors can provide current antibiotics utilization pattern.

I would like to suggest authors to add please add ATC code in front of antibiotics in table two. 

Antibiotics can be classified according to WHO AWaRe classification that is a proven method to evaluate need of antimicrobial stewardship 

Please avoid unnecessary splitting of sentences into paragraph. Only separate them when they needed to do so.

Round 2

Reviewer 1 Report

Thank you for the revision.

Minor comments;

Lines 451 - 452 on antibiotic consumption is inaccurate and discordant with the table presented. There is still quite a number of antibiotics (gram negative cover) listed in the antibiotics used, and it appears as if they may have been used for UTI. Would the authors like to comment on this?
